# LOGIT: LEARNING TO GENERATE GRADIENTS FOR FEDERATED LEARNING WITH ARBITRARY CLIENT UNAVAILABILITY

## ABSTRACT

Federated learning (FL) enables distributed clients to collaboratively train a shared model while keeping data local. However, in practice, unreliable transmissions, power constraints, and client mobility induce intermittent client unavailability, which biases gradient aggregation and impedes convergence. To address this issue, we propose LOGIT, a gradient-generation framework that learns client-specific gradient trajectories to reconstruct missing updates on the server when clients drop out. Specifically, LOGIT conditions a lightweight generator on each client's gradient history and the current-round updates from available clients, producing surrogate gradients for unavailable clients and preserving statistical diversity across participants. We further derive a tighter convergence bound and show that LOGIT converges at a rate of $\mathcal{O}(1/\sqrt{T})$, where $T$ is the number of communication rounds. Our experimental results on public datasets validate the effectiveness of LOGIT, demonstrating consistent superiority over baselines, particularly in scenarios with high data heterogeneity and client unavailability.

## 1 INTRODUCTION

Federated learning (FL) enables collaborative model training across distributed clients while keeping sensitive data local (McMahan et al., 2017). This distributed paradigm has been widely adopted in domains such as mobile computing, healthcare, and finance to enhance privacy preservation (Liu et al., 2024). In practice, however, FL encounters challenges arising from intermittent client unavailability (also known as "stragglers" problem), where unreliable transmission conditions, client mobility, and power constraints often lead to client unavailability. Such issue biases gradient aggregation, hinders convergence, and ultimately degrades overall model performance.

Existing approaches aiming at handling intermittent client unavailability primarily include asynchronous aggregation, adaptive weighting, and gradient-approximation schemes. Asynchronous FL aggregates updates as they arrive and proceeds without waiting for stragglers, thereby reducing idle time (Zhang et al., 2023). Adaptive weighting assigns aggregation weights based on heterogeneous participation rates, alleviating the bias introduced by client unavailability (Wang & Ji, 2024). Gradient-approximation methods estimate or reweight stale updates using momentum, control variates, or short-horizon extrapolation from history (Sun et al., 2024; Gu et al., 2021). In practice, however, clients may remain offline for extended and non-stationary periods, exhibiting highly heterogeneous activity patterns that induce persistent participation imbalances, bias aggregation, and slow convergence. Moreover, reconstructing missing updates solely from stale local history is fragile under distribution shifts and prolonged outages. These limitations motivate us to generate reliable surrogate updates for unavailable clients and integrate them into aggregation in a convergence-aware manner.

In this paper, we propose LOGIT, a new framework for gradient generation at the server by learning to reconstruct missing gradients through leveraging historical gradient information from the unavailability periods of clients. Specifically, LOGIT introduces a server-side gradient generation network (GGN) as a generator for each client, trained to capture the temporal dependencies in each client's gradient trajectory. When a client drops out, the generator combines its past gradients with those from available clients to generate surrogate gradients. These surrogate gradients are then incor-

porated into the aggregation process, thereby accelerating convergence and mitigating the adverse effects of client unavailability.

**Contributions.** In this paper, we summarize our contributions as follows:

- We propose LOGIT, a novel framework for gradient generation in FL, with a GGN module that is trained in an online manner to learn client-specific gradient trajectories and reconstruct missing updates on the server when clients drop out of training. To the best of our knowledge, this is the first *generative* approach in FL to address the challenge of intermittent client unavailability.

- We provide a theoretical analysis of the convergence behavior of LOGIT. Several useful insights are derived from our theoretical analysis: First, LOGIT achieves a convergence rate of $\mathcal{O}(1/\sqrt{T})$, where $T$ is the number of communication rounds. Second, the gradient generation error scales as $\mathcal{O}(\sqrt{\overline{\tau}_{\max}})$, where $\overline{\tau}_{\max}$ is the average maximum gradient staleness across all clients. This indicates that reducing the average maximum gradient staleness effectively increases the training samples available for the GGN, thereby mitigating the error. Third, under an independent and identically distributed (i.i.d) Bernoulli participation model with fixed participation probabilities, prioritizing the least reliable clients is provably more effective than simply increasing the average participation rate.

- We validate the effectiveness of LOGIT through experiments on several datasets and baselines. LOGIT outperforms baseline methods in terms of accuracy, achieving up to 4.98% improvement across all tested scenarios. Additionally, it accelerates convergence, requiring fewer communication rounds to reach the target accuracy, with a speedup of up to 1.55 times compared to the baselines. Moreover, LOGIT demonstrates superior performance under high client unavailability, as well as robustness and strong scalability with the number of clients, highlighting its effectiveness and adaptability in large-scale federated learning scenarios.

## 2 RELATED WORK

**Federated learning.** FL is introduced as an alternative to centralized training, allowing distributed clients to collaboratively optimize a shared model under server coordination without exposing raw data (McMahan et al., 2017; Gu et al., 2021; Wang & Ji, 2024; Chen et al., 2021). In each round, the server broadcasts the current global model to a subset of clients, each of which performs a few local stochastic gradient descent (SGD) steps on its private dataset and returns an update. The server then aggregates the received updates to form the new global model. This protocol reduces communication frequency while preserving privacy by exchanging only model parameters.

Federated Averaging (FedAvg) is the most widely adopted FL protocol (McMahan et al., 2017), and numerous studies have analyzed and extended it. Two central challenges are statistical heterogeneity and system heterogeneity. On the statistical side, prior analyses have characterized FL convergence under non-IID data and quantified the effect of client drift (Li et al., 2020c). To address these effects, FedProx introduces a proximal regularizer to stabilize local updates (Li et al., 2020b), while SCAFFOLD applies variance-reduction corrections to mitigate client-induced bias (Karimireddy et al., 2020). On the system side, asynchronous or buffered aggregation and client scheduling have been proposed to accommodate stragglers and heterogeneous computational capabilities (Yang et al., 2024; Nguyen et al., 2022; Wang et al., 2024; Ren et al., 2020). Despite these advances, most approaches implicitly assume that once a client is selected, its update will be received within the round or after a bounded delay—an assumption that fails under arbitrary device unavailability, which is the focus of this paper.

**Client availability.** In FL, clients often participate intermittently due to unstable connectivity, limited battery, or mobility, leading to dropouts. Recent work addresses client unavailability along three directions: (i) asynchronous or buffered aggregation, which updates the global model whenever client updates arrive while assuming delayed clients eventually respond (Yang et al., 2024; Nguyen et al., 2022; Wang et al., 2024); (ii) stale-update estimation, which caches each client's latest update and reuses or extrapolates it via momentum, short-horizon prediction, or control-variate corrections (Gu et al., 2021; Liu et al., 2020); and (iii) adaptive weighting, which adjusts aggrega-

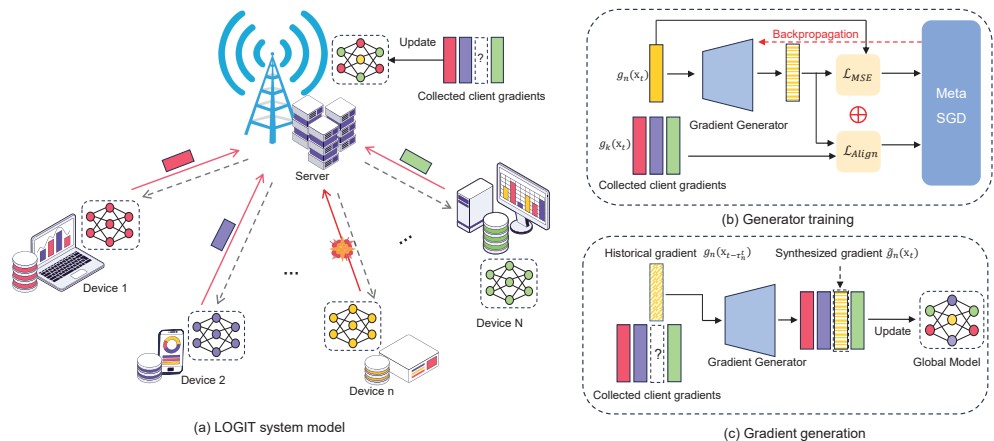

Figure 1: Overview of the proposed LOGIT framework: (a) system model, (b) generator training, and (c) gradient generation.

tion weights according to participation statistics to mitigate participation-induced bias (Wang & Ji, 2024; 2022).

Despite these advances, the above strategies are ineffective under prolonged or adversarial unavailability. Asynchronous methods address delays rather than missing updates; cached or extrapolated gradients are unreliable under distribution shift or long outages; and participation-aware weighting cannot fully correct long-term bias when missingness is non-random and correlated with data. These limitations motivate methods that generate surrogate gradients for truly missing clients and incorporate them into aggregation with calibration and convergence guarantees under stated assumptions.

**Learning to optimize.** Learning to optimize (L2O) treats optimization as a learnable procedure: instead of hand-crafted rules (e.g., SGD, Adam), a parametric policy is trained to propose update directions from past optimization signals (Andrychowicz et al., 2016). Subsequent work improves generalization and stability via meta-training (Li & Malik, 2017), hierarchical RNN optimizers (Wichrowska et al., 2017), and simple heuristics such as random scaling that speed up meta-learning (Lv et al., 2017). A central challenge is credit assignment over long unrolled trajectories; remedies include dynamic weighting for truncated backpropagation (Metz et al., 2019), gradually increasing the unroll length (Chen et al., 2020), regularization to enhance cross-task transfer (Li et al., 2020a), and bias-reduction techniques that correct truncation (Vicol et al., 2021).

L2O thus offers a practical way to design learned optimizers. Ji et al. (Ji et al., 2019) cast gradient aggregation as a learnable module and train server-side RNN/ARNN aggregators end-to-end, improving convergence and robustness to Byzantine workers. Deng et al. (Deng et al., 2025) adopt the L2O view for test-time adaptation, introducing a meta gradient generator (MGG) and a gradient-memory layer; trained in a self-supervised manner, these components distill historical gradients into parameters and synthesize reliable test-time updates from a few samples with modest computation. These results suggest that learned update rules can accelerate convergence and adapt to heterogeneous objectives beyond what fixed optimizers can offer.

## 3 PROPOSED METHODOLOGY

In this section, we introduce LOGIT, a server-side framework that generates surrogate gradients for intermittently unavailable clients by learning client-specific gradient trajectories and aligning them with the current round's set of available updates. By incorporating these surrogates into aggregation, LOGIT preserves statistical diversity and stabilizes convergence under arbitrary device unavailability.

## 3.1 PRELIMINARIES

As shown in Fig. 1, we consider a FL system with a central server and $N$ distributed clients indexed by $\mathcal{N} = \{1, 2, \ldots, N\}$. Each client $n$ holds a private dataset $\mathcal{D}_n$. The goal is to learn a global model $\mathbf{x} \in \mathbb{R}^d$ that minimizes the finite-sum objective:

$$\min_{\mathbf{x} \in \mathbb{R}^d} f(\mathbf{x}) = \frac{1}{N} \sum_{n=1}^{N} F_n(\mathbf{x}), \tag{1}$$

where $f(\mathbf{x})$ represents the global loss function, $F_n(\mathbf{x}) = \mathbb{E}_{\boldsymbol{\xi}_n \sim \mathcal{D}_n}[F_n(\mathbf{x}; \boldsymbol{\xi}_n)]$ is the local loss function of client $n$, and $\boldsymbol{\xi}_n$ denotes a mini-batch of data samples drawn from $\mathcal{D}_n$.

A standard baseline for solving (1) is FedAvg, which runs several local SGD steps per round and then averages client updates on the server. In its full-participation gradient form,

$$\mathbf{x}_t = \mathbf{x}_{t-1} - \eta \nabla F(\mathbf{x}_{t-1}) = \mathbf{x}_{t-1} - \eta \frac{1}{N} \sum_{n=1}^{N} \nabla F_n(\mathbf{x}_{t-1}), \tag{2}$$

where $\eta > 0$ is the learning rate, $\mathbf{x}_t$ is the global parameter at round $t$, $\nabla F(\mathbf{x}_{t-1})$ is the global gradient, and $\nabla F_n(\mathbf{x}_{t-1})$ is the local gradient of client $n$.

In practice, network volatility, battery limits, and mobility frequently break full participation, biasing the aggregated direction and slowing convergence. This motivates reconstructing missing updates in a principled manner rather than discarding them.

## 3.2 GRADIENT GENERATION NETWORK (GGN)

We introduce the Gradient Generation Network (GGN), a key component of LOGIT implemented on the server side, which generates surrogate gradients for intermittently unavailable clients. By conditioning on client-specific gradient history and the updates from available clients, the GGN produces gradients that preserve statistical diversity and enforce knowledge alignment with the global gradient descent direction estimated from the received updates.

**Training objective.** Let $g_n(\mathbf{x}_t)$ denote denote the stochastic gradient accumulated by client $n$ at round $t$. Denote the available-client set by $\mathcal{A}_t$ with $|\mathcal{A}_t| = K$. As shown in Fig. 1 (b), for client $n \in \mathcal{A}_t$, the GGN takes as input both the client-specific gradient $g_n(\mathbf{x}_t)$ and the updates from other available clients $g_n(\mathbf{x}_t), k \in \mathcal{A}_t \setminus n$. These inputs are used to learn the client's gradient trajectory, capturing the temporal dependencies and dynamics of the client's gradients over time, while also estimating the global gradient descent direction from the received updates. The GGN then embeds this information into its network, generating surrogate gradients that can be used to approximate the missing gradients when a client becomes unavailable. Specifically, let $f_{\mathrm{GGN}}(\cdot; \boldsymbol{\theta}_n)$ denote the gradient generator of client $n$, with learnable parameters $\boldsymbol{\theta}_n$. We train it by minimizing

$$\mathcal{L}_{\mathrm{GNN}}(g_n(\mathbf{x}_t); \boldsymbol{\theta}_n^{t-1}) = \underbrace{(1 - \lambda_n) \left\| f_{\mathrm{GGN}}(g_n(\mathbf{x}_t); \boldsymbol{\theta}_n^{t-1}) - g_n(\mathbf{x}_t) \right\|^2}_{\mathcal{L}_{\mathrm{MSE}}}$$

$$\underbrace{- \frac{\lambda_n}{K-1} \sum_{k \in \mathcal{A}_t \setminus n} \left\langle f_{\mathrm{GGN}}(g_n(\mathbf{x}_t); \boldsymbol{\theta}_n^{t-1}), g_k(\mathbf{x}_t) \right\rangle}_{\mathcal{L}_{\mathrm{Align}}}, \tag{3}$$

where $f_{\mathrm{GGN}}(g_n(\mathbf{x}_t); \boldsymbol{\theta}_n^{t-1})$ denotes the output of GGN, $\| \cdot \|$ denotes the Euclidean norm, $\langle \cdot \rangle$ represents the cosine similarity. denotes the cosine similarity. The first term corresponds to a self-reconstruction loss $\mathcal{L}_{\mathrm{MSE}}$ that captures client-specific dynamics, while the second corresponds to a cross-client alignment loss $\mathcal{L}_{\mathrm{Align}}$ that encourages the generator to align with the population geometry of the current round. The weight $\lambda_n \in [0, 1]$ balances between local fidelity and global consistency.

**Parameter update.** GGN parameters are updated as

$$\boldsymbol{\theta}_n^t = \boldsymbol{\theta}_n^{t-1} - \eta_{\mathrm{GGN}} \nabla_{\boldsymbol{\theta}} \mathcal{L}_{\mathrm{GGN}}(g_n(\mathbf{x}_t); \boldsymbol{\theta}_n^{t-1}), \tag{4}$$

where $\eta_{\text{GNN}}$ denotes the learning rate for the GGN update. In practice, we apply gradient clipping to $\nabla_{\boldsymbol{\theta}} \mathcal{L}_{\text{GGN}}$ and an weight decay to $\boldsymbol{\theta}_n$ to stabilize training.

**Gradient generation.** As shown in Fig. 1 (c), when client $n$ experiences a transmission failure in round $t$, the server generates a surrogate gradient by mapping its most recent stored gradient to the current round:

$$\tilde{g}_n(\mathbf{x}_t) = f_{\text{GGN}}(g_n(\mathbf{x}_{t-\tau_n^t}), \boldsymbol{\theta}_n^t), \tag{5}$$

where $\tau_n^t \in \mathbb{N}$ denotes the staleness of client $n'$ latest received update, $\tilde{g}_n(\mathbf{x}_t)$ is the surrogate gradient generated by the GGN for client $n$ at time step $t$.

**Client-specific coordinatewise GGN.** Generating gradients for high-dimensional models with an enormous number of parameters is computationally expensive. We therefore instantiate a separate generator for each client $n$ and adopt a coordinatewise parameterization. Concretely, the same small network $\phi$ with parameters $\boldsymbol{\theta}_n$ is applied to every coordinate,

$$[f_{\text{GGN}}(g)]_j = \phi(g_j; \boldsymbol{\theta}_n), j = 1, 2, \cdots, d, \tag{6}$$

sharing weights across coordinates within client $n$ This coordinatewise design yields several benefits: (i) linear scaling in the dimension with per-round server compute $\mathcal{O}(Kd)$; (ii) invariance to parameter ordering since the same rule is applied independently to each coordinate; (iii) low compute and memory via a shared lightweight network; and (iv) no extra client-side communication beyond standard FL traffic (Andrychowicz et al., 2016).

## 3.3 FL PROTOCOL

At the beginning of round $t$, the server broadcasts the current global model $\mathbf{x}_t$ to all clients. Each client $n$ initializes $\mathbf{x}_{t,0}^n = \mathbf{x}_t$ and performs $I$ local SGD steps with learning rate $\eta$:

$$\mathbf{x}_{t,i}^n = \mathbf{x}_{t,i-1}^n - \eta g_n(\mathbf{x}_{t,i-1}^n), \quad i = 1, \cdots, I, \tag{7}$$

where $g_n(\cdot)$ denotes the stochastic gradient computed on client on client $n$. The accumulated local gradient used for aggregation is $g_n(\mathbf{x}_t) = \sum_{i=1}^{I-1} g_n(\mathbf{x}_{t,i-1}^n)$.

After local updates, clients attempt to upload their gradients to the server. Let $\mathbb{I}_n^t \in \{0, 1\}$ be an indicator variable that equals 1 if the gradient of client $n$ is received in round $t$. For clients that are unavailable, the server generates surrogate gradients via (5). The global model is then updated by aggregating the received and surrogate gradients:

$$\mathbf{x}_{t+1} = \mathbf{x}_t - \eta \frac{1}{N} \sum_{n=1}^{N} \left[ \mathbb{I}_n^t g_n(\mathbf{x}_t) + (1 - \mathbb{I}_n^t) \tilde{g}_n^t(\mathbf{x}_t) \right]. \tag{8}$$

Note that weighted aggregation with client priors can be used as a drop-in replacement; we report uniform weighting for clarity. Finally, the server refreshes the cache of last-received gradients for the available set $\mathcal{A}_t = \{n : \mathbb{I}_n^t = 1\}$:

$$g_n(\mathbf{x}_{t-\tau_n^t}) \leftarrow g_n(\mathbf{x}_t), \forall n \in \mathcal{A}(t). \tag{9}$$

Unless otherwise specified, gradients are $\ell_2$-normalized before being fed to the GGN (for both training and synthesis), while raw, unnormalized vectors are used in the aggregation of (8). The complete procedure is summarized in Algorithm 1.

## 4 CONVERGENCE ANALYSIS

In this section, we present a theoretical convergence analysis of the proposed FL scheme.

**Assumption 1.** *The local loss function is Lipschitz continuous with a positive constant $L$ for each client $n \in \mathcal{N}$, i.e., $\forall \mathbf{x}, \mathbf{y} \in \mathbb{R}^d, \|\nabla F_n(\mathbf{x}) - \nabla F_n(\mathbf{y})\| \leq L \|\mathbf{x} - \mathbf{y}\|$.*

**Assumption 2.** *The stochastic gradients for each client $n$ is unbiased, and its variance is bounded, i.e., $\mathbb{E}[g_n(\mathbf{x})] = \nabla F_n(\mathbf{x}), \mathbb{E}[\|g_n(\mathbf{x}) - \nabla F_n(\mathbf{x})\|^2] \leq \sigma_1^2, \forall \mathbf{x}$.*

**Assumption 3.** *The surrogate gradient computed on a mini-batch sampled at client $n$ is unbiased, and its variance is bounded, i.e., $\mathbb{E}[\tilde{g}_n(\mathbf{x})] = \nabla \tilde{F}_n(\mathbf{x}), \mathbb{E}[\|\tilde{g}_n(\mathbf{x}) - \nabla \tilde{F}_n(\mathbf{x})\|^2] \leq \sigma_2^2, \forall \mathbf{x}$.*

---

**Algorithm 1** LOGIT: Surrogate Gradient Generation under Arbitrary Client Unavailability

---

**Require:** Client set $\mathcal{N}$, rounds $T$, local steps $I$, learning rates $\eta, \eta_{\text{GGN}}$, initial model $\mathbf{x}_0$, GGN parameters $\{\boldsymbol{\theta}_n^0\}, n \in \mathcal{N}$

**Output:** Global model $\mathbf{x}_T$

1: **for** $t = 1$ to $T$ **do**
2:     **Server** broadcasts $\mathbf{x}_t$ to all clients.
3:     **for all** $n \in \mathcal{N}$ **(in parallel) do**
4:         $\mathbf{x}_{t,0}^n \leftarrow \mathbf{x}_t$
5:         **for** $i = 1$ to $I$ **do**
6:             $\mathbf{x}_{t,i}^n \leftarrow \mathbf{x}_{t,i-1}^n - \eta\, g_n(\mathbf{x}_{t,i-1}^n)$
7:         **end for**
8:         Compute local gradient $g_n(\mathbf{x}_t) \leftarrow \sum_{i=0}^{I-1} g_n(\mathbf{x}_{t,i}^n)$.
9:         With probability $p_n^t$, upload $g_n(\mathbf{x}_t)$ to server (set $\mathbb{I}_n^t=1$); otherwise no upload ($\mathbb{I}_n^t=0$).
10:    **end for**
11:    Let $\mathcal{A}_t = \{n \in \mathcal{N} : \mathbb{I}_n^t = 1\}$ be the set of active clients.
12:    **Train GGN:**
13:    **for all** $n \in \mathcal{A}_t$ **do**
14:        $\boldsymbol{\theta}_n^t \leftarrow \boldsymbol{\theta}_n^{t-1} - \eta_{\text{GGN}} \nabla_{\boldsymbol{\theta}} L_{\text{GGN}}\big(g_n(\mathbf{x}_t); \boldsymbol{\theta}_n^{t-1}\big)$
15:    **end for**
16:    **Generate surrogate gradients:**
17:    **for all** $n \in \mathcal{N} \setminus \mathcal{A}_t$ **do**
18:        Obtain staleness $\tau_n^t$ and set $\tilde{g}_n(\mathbf{x}_t) \leftarrow f_{\text{GGN}}\big(g_n(\mathbf{x}_{t-\tau_n^t}); \boldsymbol{\theta}_n^t\big)$
19:    **end for**
20:    **Aggregate:**
21:    $\mathbf{x}_{t+1} \leftarrow \mathbf{x}_t - \eta \frac{1}{|\mathcal{N}|} \sum_{n \in \mathcal{N}} \big(\mathbb{I}_n^t g_n(\mathbf{x}_t) + (1 - \mathbb{I}_n^t)\tilde{g}_n(\mathbf{x}_t)\big)$
22: **end for**

---

**Assumption 4.** *The divergence between local and global gradients is bounded, i.e.,* $\mathbb{E}[\|\nabla F_n(\mathbf{x}) - \nabla f(\mathbf{x})\|^2] \leq \sigma_3^2, \forall \mathbf{x}$.

Assumptions 1, 2, and 4 are widely used in the convergence analysis literature in FL (Yang & Liu, 2025; Liu et al., 2025; Wang & Ji, 2022; Gu et al., 2021), while Assumption 3 is introduced to control the variance of local stochastic estimates induced by the gradient generation error (Nair et al., 2025).

**Theorem 1.** *When Assumptions 1–4 hold, with learning rate* $\eta = \min\{\frac{1}{\sqrt{IT}}, \frac{1}{2\sqrt{30}LI}\}$, *the convergence rate of the proposed method satisfies*

$$
\frac{1}{T} \sum_{t=1}^{T} \mathbb{E}\left[\|\nabla f(\mathbf{x_t})\|^2\right]
$$

$$
\leq \mathcal{O}\left( \frac{(\mathbb{E}\left[f(\mathbf{x_0}) - f(\mathbf{x^*})\right])}{\sqrt{IT}} + \frac{L(\sigma_1^2 \overline{p} + \sigma_2^2 \overline{q})}{N\sqrt{IT}} + \frac{L^2(\sigma_1^2 + I\sigma_3^2)}{T} + \overline{\epsilon}^* + \frac{\sqrt{\overline{\tau}_{\max}(1 - \overline{p}_{\min})}}{\sqrt{T}} \right),
$$

$$
\tag{10}
$$

*where* $\overline{p} = \frac{1}{TN} \sum_{t=1}^{T} \sum_{n=1}^{N} p_n^t$ *denotes the average transmission success probability,* $\overline{q} = 1 - \overline{p}$, $\overline{\epsilon}^* = \frac{1}{TN} \sum_{t=1}^{T} \sum_{n=1}^{N} (1 - p_n^t)\epsilon_{n,t}^*$ *is the approximation error due to the limitations in the gradient generator's ability to approximate the true gradients, and* $\overline{\tau}_{\max} = \frac{1}{N} \sum_{n=1}^{N} \tau_{n,\max}$ *is the average maximum gradient staleness across all clients, where* $\tau_{n,\max} = \max\{\tau_n^t\}, \forall t$. *Additionally,* $\overline{p}_{\min} = \frac{1}{N} \sum_{n=1}^{N} p_{n,\min}$ *is the average minimum transmission success probability across all clients, where* $p_{n,\min} = \min\{p_n^t\}, \forall t$.

Theorem 1 shows that the proposed method achieves a convergence rate of $\mathcal{O}(1/\sqrt{T})$ for the averaged gradient norm. The convergence behavior depends on client participation probability, gradient staleness, local epochs, and generator accuracy. The variance term (the second term on the right-hand side) scales with the mixture noise $(\sigma_1^2 \overline{p} + \sigma_2^2 \overline{q})$ and improves linearly with larger average participation probability. The term $\mathcal{O}\left( \sqrt{\overline{\tau}_{\max}(1 - \overline{p}_{\min})}/\sqrt{T} \right)$ reflects the interaction between

the worst-case staleness and the least reliable clients, suggesting the need to bound the maximum allowable $\tau_{n,\max}$ and prioritize the stragglers with the minimum transmission success probability $p_{n,\min}$. The surrogate-gradient bias contributes an additive error floor $\bar{\epsilon}^*$, which can be reduced by increasing the generator's capacity (e.g., model size) or providing more training samples. Finally, local computation introduces a trade-off: increasing $I$ reduces the $\frac{1}{TI}$ optimization term but also amplifies heterogeneity drift and tightens the stepsize bound. Thus, a moderate choice of $I$ with $\eta \approx \frac{1}{\sqrt{TI}}$ is preferred. A full proof of Theorem 1 is provided in Appendix A.1.

**Theorem 2.** *(Special case): Suppose each client follows an i.i.d. Bernoulli participation model with probabilities $\{p_n\}$, i.e., $p_n^t = p_n, \forall t$, we have*

$$\frac{1}{T} \sum_{t=1}^{T} \mathbb{E}\left[\|\nabla f(\mathbf{x_t})\|^2\right]$$

$$\leq \mathcal{O}\left(\frac{(\mathbb{E}\left[f(\mathbf{x_0}) - f(\mathbf{x}^*)\right])}{\sqrt{IT}} + \frac{L(\sigma_1^2 \overline{p} + \sigma_2^2 \overline{q})}{N\sqrt{IT}} + \frac{L^2(\sigma_1^2 + I\sigma_3^2)}{T} + \bar{\epsilon}^* + \frac{(1 - p_{\min})^{\frac{3}{2}}}{\sqrt{T}p_{\min}}\right). \quad (11)$$

*where $\overline{p} = \frac{1}{TN} \sum_{t=1}^{T} \sum_{n=1}^{N} p_n$ denotes the average transmission success probability, $\overline{q} = 1 - \overline{p}$, $\bar{\epsilon}^* = \frac{1}{TN} \sum_{t=1}^{T} \sum_{n=1}^{N} (1 - p_n)\epsilon_{n,t}^*$ is the approximation error due to the limitations in the gradient generator's ability to approximate the true gradients, and $p_{n,\min} = \min\{p_n\}, \forall n$.*

Compared with Theorem 1, the i.i.d. Bernoulli setting replaces the unknown worst-run staleness factor $\overline{\tau}_{\max}$ with a clean participation term $(O)\left((1 - p_{\min})^{\frac{3}{2}}/\sqrt{p_{\min}}\right)$. This removes the need to assume a fixed $\tau_{\max}$ and makes the dependence on the least reliable client explicit. As $p_{\min} \to 1$, the extra staleness term vanishes and the standard $\mathcal{O}(1/\sqrt{T})$ behavior is recovered; when $p_{\min}$ is small, that term dominates and becomes the bottleneck. Therefore, raising $p_{\min}$ (prioritizing the least reliable client) is typically more effective than merely increasing the average participation rate. A full proof of Theorem 2 is provided in Appendix A.2.

## 5 EXPERIMENTS

In this section, we conduct numerical experiments on public datasets to evaluate the performance of LOGIT.

**Implementation details.**

In the experiments, we set the number of clients to $N = 10$. The FL learning rate is $\eta = 0.01$, and the GGN learning rate is $\eta_{\text{GGN}} = 0.001$. The local mini-batch size is fixed at 32. Client availability follows an independent Bernoulli process with success probability $p_n^t$. Unless otherwise stated, $p_n^t$ is set to 0.5 for all client. We conduct experiments on three public computer vision datasets, CIFAR-10, CIFAR-100, and IMAGENETTE. To ensure fairness, each client is assigned the same number of samples with $|\mathcal{D}_n| = 2048$. Data heterogeneity is simulated by drawing client partitions from a Dirichlet distribution $\text{Dir}(\alpha)$, where $\alpha$ is the Dirichlet parameter. We consider $\alpha \in 0.1, 0.3$, with smaller $\alpha$ corresponding to higher heterogeneity. We adopt ResNet-18 as the backbone model for distributed training (He et al., 2016). The total number of training rounds $T$ is set to 600. For the GGN, we employ a single-layer LSTM that takes coordinate-wise historical gradients and the previous hidden state as inputs, and outputs surrogate gradients (Andrychowicz et al., 2016).

**Baselines.**

To evaluate the effectiveness of the proposed method, we compare it against the following baselines:

- FedAvg: FedAvg (McMahan et al., 2017) aggregates only the received updates from available clients and directly discards the updates from unavailable clients.
- MIFA: MIFA (Gu et al., 2021; Xiang et al., 2024) avoids waiting for dropped-out clients and reuses the most recently memorized updates as surrogates when a client is unavailable.
- WS: In WS (Zhao et al., 2025), if a client experiences a transmission failure, the server computes the client's gradient as a weighted sum of the most recent local gradient and the last global gradient. This method combines local and global information to approximate

Table 1: Top-1 test accuracy (%).

| Method | CIFAR-10 | | CIFAR-100 | | IMAGENETTE | |
|---|---|---|---|---|---|---|
| | $\alpha = 0.1$ | $\alpha = 0.3$ | $\alpha = 0.1$ | $\alpha = 0.3$ | $\alpha = 0.1$ | $\alpha = 0.3$ |
| FedAvg | 77.67 | 88.73 | 57.07 | 65.62 | 70.42 | 82.80 |
| MIFA | 78.05 | 88.67 | 57.55 | 66.13 | 70.01 | 82.69 |
| WS | 77.75 | 88.68 | 57.45 | 66.17 | 69.88 | 82.77 |
| LOGIT (Ours) | **82.11** | **89.56** | **58.65** | **66.94** | **74.78** | **84.26** |

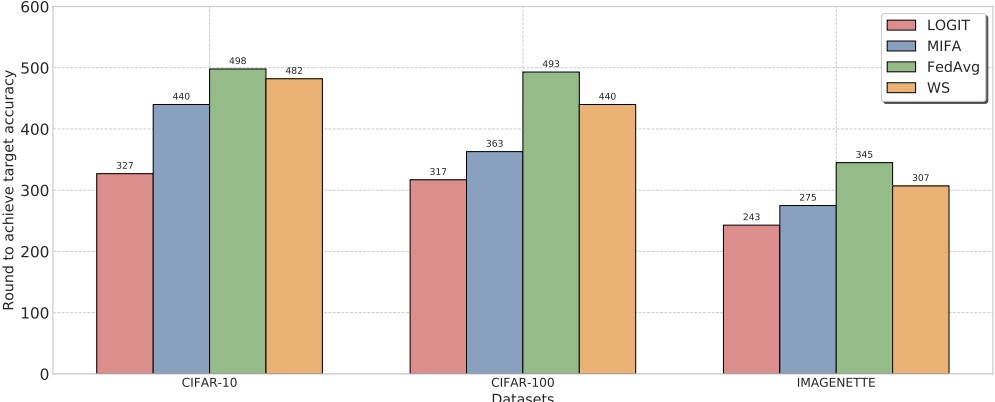

Figure 2: Total communication rounds required to achieve the target accuracy of 75% on CIFAR-10, 55% on CIFAR-100, and 65% on IMAGENETTE.

the missing updates. To ensure fairness, the weighting coefficient for each client $n$ is set to be consistent with $\lambda_n$ in the proposed method.

**Result and discussion.**

We evaluate the performance of the proposed method across different datasets and under varying degrees of data heterogeneity. For robustness, all experiments are conducted with multiple random seeds, and we report the average top-1 accuracy across seeds in Table 1. The results show that the proposed method consistently outperforms the baselines, especially in the presence of extreme data heterogeneity. This accuracy gain can be attributed to two main factors. First, the method generates surrogate gradients for unavailable clients, thereby preserving the statistical diversity of the participant set and effectively mitigating aggregation bias. Second, during GGN training, an alignment mechanism leverages updates from available clients to guide the gradient generation process, ensuring consistency with global knowledge. In addition, we compare the communication overhead of our method with the baselines at the same target accuracy in Fig. 2. The results show that the proposed method achieves the target accuracy with fewer communication rounds, indicating that it is more communication-efficient and converges faster than the baselines. This advantage arises because our method compensates for missing global statistics by generating gradients for unavailable clients, which is fundamentally different from and more efficient than tracking historical updates as in MIFA and WS.

**Ablation study.**

In this subsection, we present an ablation study on the transmission success probability, maximum staleness, and the number of clients. Fig. 3(a) shows that the average top-1 accuracy increases as the transmission success probability improves, with our method consistently achieving higher accuracy than the baselines. This improvement arises because more available clients contribute richer information to the FL model and provide additional training samples for the GGN, enabling it to better capture client gradient trajectories and thereby reduce the synthesis error. Fig. 3(b) illustrates that the average top-1 accuracy decreases as gradient staleness increases. Nevertheless, our method maintains a clear advantage over the baselines. This robustness stems from the gradient alignment mechanism, which allows the GGN to acquire global knowledge in an online manner

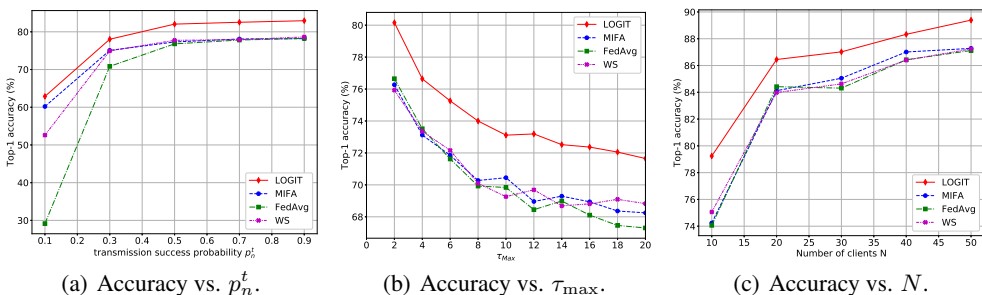

(a) Accuracy vs. $p_n^t$.      (b) Accuracy vs. $\tau_{\max}$.      (c) Accuracy vs. $N$.

Figure 3: Ablation study on transmission success probability, maximum staleness, and the number of clients.

during training and ensures that the generative gradients remain aligned with the global knowledge, thereby guiding the generation of more reliable surrogate gradients. Finally, Fig. 3(c) shows that the average top-1 accuracy of our method increases with the number of clients. This trend is explained by the fact that more clients provide more informative gradient embeddings. Moreover, our method achieves higher accuracy than the baselines across all client scales, demonstrating both its efficiency and scalability.

## 6 CONCLUSION

In this work, we propose LOGIT, a novel framework for gradient generation in FL, specifically addressing the challenge of intermittent client unavailability. By leveraging historical gradient information and current updates from available clients, LOGIT generates surrogate gradients that are incorporated into the aggregation process, preserving statistical diversity and accelerating convergence. We further derive a convergence bound for the proposed method. Our experimental results on public benchmarks validate the effectiveness of LOGIT, consistently outperforming strong baselines, especially in scenarios with high data heterogeneity and client unavailability. Furthermore, LOGIT shows improved communication efficiency, reducing the number of communication rounds required to reach the target accuracy. These findings underscore the scalability, robustness, and practical applicability of the proposed method in real-world FL settings.

## LLM USAGE STATEMENT

This work was conceived, designed, and executed by the authors. A large language model was used to copy-edit author-written drafts (grammar, phrasing, and minor formatting).

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

## A  APPENDIX

### A.1  PROOFS OF THEOREM 1

According to (8), in each round, the server updates the global model by aggregating the received and surrogate gradients as follows:

$$\mathbf{x_{t+1}} = \mathbf{x_t} - \eta \frac{1}{N} \sum_{n=1}^{N} \left[ \mathbb{I}_n^t g_n(\mathbf{x_t}) + (1 - \mathbb{I}_n^t) \tilde{g}_n(\mathbf{x_t}) \right], \tag{12}$$

where $\mathbb{I}_n^t$ is an indicator function, with $\mathbb{I}_n^t = 1$ indicating that client $n$ transmits successfully and $\mathbb{I}_n^t = 0$ indicating that device $n$ experiences a transmission failure, $g_n(\mathbf{x_t}) = \sum_{i=1}^{I-1} g_n(\mathbf{x}_{t,i-1}^n)$, and $\tilde{g}_n(\mathbf{x_t})$ is the surrogate gradient generated by the GGN for client $n$ at round $t$.

According to Assumption 1, we have

$$\mathbb{E}\left[f(\mathbf{x_{t+1}})\right] \le f(\mathbf{x_t}) - \eta \mathbb{E}\left[\langle \nabla f(\mathbf{x_t}), \mathbf{x_{t+1}} - \mathbf{x_t}\rangle\right] + \frac{L\eta^2}{2}\mathbb{E}\left[\|\mathbf{x_{t+1}} - \mathbf{x_t}\|^2\right]$$

$$= f(\mathbf{x_t}) - \eta \underbrace{\mathbb{E}\left[\left\langle \nabla f(\mathbf{x_t}), \frac{1}{N}\sum_{n=1}^{N}\left(\mathbb{I}_n^t g_n(\mathbf{x_t}) + (1 - \mathbb{I}_n^t)\tilde{g}_n(\mathbf{x_t})\right)\right\rangle\right]}_{A}$$

$$+ \frac{L\eta^2}{2}\underbrace{\mathbb{E}\left[\left\|\frac{1}{N}\sum_{n=1}^{N}\left(\mathbb{I}_n^t g_n(\mathbf{x_t}) + (1 - \mathbb{I}_n^t)\tilde{g}_n(\mathbf{x_t})\right)\right\|^2\right]}_{B}, \tag{13}$$

where the expectation $\mathbb{E}[\cdot]$ is taken over all the randomness up to the $t$-th round (Nair et al., 2025).

According to Assumptions 2 and 3, we can bound the term $A$ as follows:

$$A = \left\langle \nabla f(\mathbf{x_t}), -\frac{\eta}{N}\mathbb{E}\left[IN\nabla f(\mathbf{x_t}) + \sum_{n=1}^{N}\left(\mathbb{I}_n^t g_n(\mathbf{x_t}) + (1 - \mathbb{I}_n^t)\tilde{g}_n(\mathbf{x_t})\right) - IN\nabla f(\mathbf{x_t})\right]\right\rangle$$

$$= \left\langle \nabla f(\mathbf{x_t}),\right.$$

$$\left. -\frac{\eta}{N}\mathbb{E}\left[IN\nabla f(\mathbf{x_t}) - IN\nabla f(\mathbf{x_t}) + \sum_{n=1}^{N}\sum_{i=0}^{I-1}\left(\mathbb{I}_n^t \nabla F_n(\mathbf{x}_{t,i}^n) + (1 - \mathbb{I}_n^t)\nabla \tilde{F}_n(\mathbf{x}_{t,i}^n)\right)\right]\right\rangle$$

$$= -\eta I\|\nabla f(\mathbf{x_t})\|^2 + \mathbb{E}\left[\left\langle \sqrt{I\eta}\nabla f(\mathbf{x_t}), -\frac{\sqrt{\eta}}{N\sqrt{I}}\sum_{n=1}^{N}\sum_{i=0}^{I-1}\left\{\mathbb{I}_n^t\left(\nabla F_n(\mathbf{x}_{t,i}^n) - \nabla F_n(\mathbf{x_t})\right)\right.\right.\right.$$

$$\left.\left.\left. + (1 - \mathbb{I}_n^t)\left(\nabla \tilde{F}_n(\mathbf{x}_{t,i}^n) - \nabla F_n(\mathbf{x_t})\right)\right\}\right\rangle\right]$$

$$\overset{(a)}{=} -\frac{\eta I}{2}\|\nabla f(\mathbf{x_t})\|^2 - \frac{\eta}{2N^2 I}\mathbb{E}\left[\left\|\sum_{n=1}^{N}\sum_{i=0}^{I-1}\left\{\mathbb{I}_n^t \nabla F_n(\mathbf{x}_{t,i}^n) + (1 - \mathbb{I}_n^t)\nabla \tilde{F}_n(\mathbf{x}_{t,i}^n)\right\}\right\|^2\right]$$

$$+ \frac{\eta}{2N^2 I}\mathbb{E}\left[\left\|\sum_{n=1}^{N}\sum_{i=0}^{I-1}\left\{\mathbb{I}_n^t\left(\nabla F_n(\mathbf{x}_{t,i}^n) - \nabla F_n(\mathbf{x_t})\right) + (1 - \mathbb{I}_n^t)\left(\nabla \tilde{F}_n(\mathbf{x}_{t,i}^n) - \nabla F_n(\mathbf{x_t})\right)\right\}\right\|^2\right]$$

$$= -\frac{\eta I}{2}\|\nabla f(\mathbf{x_t})\|^2 - \frac{\eta}{2N^2 I}\mathbb{E}\left[\left\|\sum_{n=1}^{N}\sum_{i=0}^{I-1}\left\{\mathbb{I}_n^t \nabla F_n(\mathbf{x}_{t,i}^n) + (1 - \mathbb{I}_n^t)\nabla \tilde{F}_n(\mathbf{x}_{t,i}^n)\right\}\right\|^2\right]$$

$$+ \frac{\eta}{2N^2 I}\mathbb{E}\left[\left\|\sum_{n=1}^{N}\sum_{i=0}^{I-1}\left\{\mathbb{I}_n^t\left(\nabla F_n(\mathbf{x}_{t,i}^n) - \nabla F_n(\mathbf{x_t})\right)\right.\right.\right.$$

$$\left.\left.\left. + (1 - \mathbb{I}_n^t)\left(\nabla \tilde{F}_n(\mathbf{x}_{t,i}^n) - \nabla F_n(\mathbf{x}_{t,i}^n) + \nabla F_n(\mathbf{x}_{t,i}^n) - \nabla F_n(\mathbf{x_t})\right)\right\}\right\|^2\right]$$

$$
= -\frac{\eta I}{2}\|\nabla f(\mathbf{x}_t)\|^2 - \frac{\eta}{2N^2 I}\mathbb{E}\left[\left\|\sum_{n=1}^{N}\sum_{i=0}^{I-1}\left\{\mathbb{I}_n^t\nabla F_n(\mathbf{x}_{t,i}^{n}) + (1-\mathbb{I}_n^t)\nabla\tilde{F}_n(\mathbf{x}_{t,i}^{n})\right\}\right\|^2\right]
$$

$$
+ \frac{\eta}{2N^2 I}\mathbb{E}\left[\left\|\sum_{n=1}^{N}\sum_{i=0}^{I-1}\left\{\left(\nabla F_n(\mathbf{x}_{t,i}^{n}) - \nabla F_n(\mathbf{x}_t)\right) + (1-\mathbb{I}_n^t)\left(\nabla\tilde{F}_n(\mathbf{x}_{t,i}^{n}) - \nabla F_n(\mathbf{x}_{t,i}^{n})\right)\right\}\right\|^2\right],
$$

$$(14)$$

where equality $(a)$ follows from the property $\langle\mathbf{z_1},\mathbf{z_2}\rangle = \frac{\|\mathbf{z_1}\|^2}{2} + \frac{\|\mathbf{z_2}\|^2}{2} - \frac{\|\mathbf{z_1}-\mathbf{z_2}\|^2}{2}$, $I$ denotes the number of local epochs.

Next, we bound the term $B$ as follows.

$$
B = \frac{L\eta^2}{2}\mathbb{E}\left[\left\|\frac{1}{N}\sum_{n=1}^{N}\left(\mathbb{I}_n^t g_n(\mathbf{x}_t) + (1-\mathbb{I}_n^t)\tilde{g}_n(\mathbf{x}_t)\right)\right\|^2\right]
$$

$$
= \frac{L\eta^2}{2N^2}\mathbb{E}\left[\left\|\sum_{n=1}^{N}\sum_{i=0}^{I-1}\left\{\mathbb{I}_n^t\left(g_n(\mathbf{x}_{t,i}^{n}) - \nabla F_n(\mathbf{x}_{t,i}^{n}) + \nabla F_n(\mathbf{x}_{t,i}^{n})\right)\right.\right.\right.
$$

$$
\left.\left.\left. + (1-\mathbb{I}_n^t)\left(\tilde{g}_n(\mathbf{x}_t) - \nabla\tilde{F}_n(\mathbf{x}_{t,i}^{n}) + \nabla\tilde{F}_n(\mathbf{x}_{t,i}^{n})\right)\right\}\right\|^2\right]
$$

$$
\overset{(b)}{=} \frac{L\eta^2}{2N^2}\mathbb{E}\left[\left\|\sum_{n=1}^{N}\sum_{i=0}^{I-1}\left\{\mathbb{I}_n^t\left(g_n(\mathbf{x}_{t,i}^{n}) - \nabla F_n(\mathbf{x}_{t,i}^{n})\right) + (1-\mathbb{I}_n^t)\left(\tilde{g}_n(\mathbf{x}_t) - \nabla\tilde{F}_n(\mathbf{x}_{t,i}^{n})\right)\right\}\right\|^2\right]
$$

$$
+ \frac{\eta}{2N^2}\mathbb{E}\left[\left\|\sum_{n=1}^{N}\sum_{i=0}^{I-1}\left\{\mathbb{I}_n^t\nabla F_n(\mathbf{x}_{t,i}^{n}) + (1-\mathbb{I}_n^t)\nabla\tilde{F}_n(\mathbf{x}_{t,i}^{n})\right\}\right\|^2\right]
$$

$$
\overset{(c)}{\le} \frac{LI\eta^2\sigma_1^2}{2N^2}\sum_{n=1}^{N}p_n^t + \frac{LI\eta^2\sigma_2^2}{2N^2}\sum_{n=1}^{N}(1-p_n^t)
$$

$$
+ \frac{\eta}{2N^2}\mathbb{E}\left[\left\|\sum_{n=1}^{N}\sum_{i=0}^{I-1}\left\{\mathbb{I}_n^t\nabla F_n(\mathbf{x}_{t,i}^{n}) + (1-\mathbb{I}_n^t)\nabla\tilde{F}_n(\mathbf{x}_{t,i}^{n})\right\}\right\|^2\right],
$$

$$(15)$$

where equality $(b)$ follows from Assumption 2, and inequality $(c)$ uses $\mathbb{E}[\mathbb{I}_n^t] = p_n^t$. Note that in this paper, we assume that the participation indicators $\mathbb{I}_n^t$ are independent across clients, the stochastic gradients are independent across different clients $n$ and local epochs $i$, and that $\mathbb{I}_n^t$ is independent of the stochastic gradients (Wang & Ji, 2024).

Substituting (14) and (15) into (13), and letting $\eta \le \frac{1}{LI}$, we obtain

$$
\mathbb{E}\left[f(\mathbf{x_{t+1}})\right]
$$

$$
\le f(\mathbf{x_t}) - \frac{\eta I}{2}\|\nabla f(\mathbf{x_t})\|^2 + \frac{LI\eta^2\sigma_1^2}{2N^2}\sum_{n=1}^{N}p_n^t + \frac{LI\eta^2\sigma_2^2}{2N^2}\sum_{n=1}^{N}(1-p_n^t)
$$

$$
+ \frac{\eta}{2N^2 I}\mathbb{E}\left[\left\|\sum_{n=1}^{N}\sum_{i=0}^{I-1}\left\{\mathbb{I}_n^t\left(\nabla F_n(\mathbf{x}_{t,i}^{n}) - \nabla F_n(\mathbf{x_t})\right) + (1-\mathbb{I}_n^t)\left(\nabla\tilde{F}_n(\mathbf{x}_{t,i}^{n}) - \nabla F_n(\mathbf{x_t})\right)\right\}\right\|^2\right]
$$

$$
- \frac{\eta(1-L\eta I)}{2N^2 I}\mathbb{E}\left[\left\|\sum_{n=1}^{N}\sum_{i=0}^{I-1}\left\{\mathbb{I}_n^t\nabla F_n(\mathbf{x}_{t,i}^{n}) + (1-\mathbb{I}_n^t)\nabla\tilde{F}_n(\mathbf{x}_{t,i}^{n})\right\}\right\|^2\right]
$$

$$
\le f(\mathbf{x_t}) - \frac{\eta I}{2}\|\nabla f(\mathbf{x_t})\|^2 + \frac{LI\eta^2\sigma_1^2}{2N^2}\sum_{n=1}^{N}p_n^t + \frac{LI\eta^2\sigma_2^2}{2N^2}\sum_{n=1}^{N}(1-p_n^t)
$$

$$+ \frac{\eta}{2N^2I} \mathbb{E} \left[ \underbrace{ \left\| \sum_{n=1}^{N} \sum_{i=0}^{I-1} \left\{ \left( \nabla F_n(\mathbf{x}_{t,i}^n) - \nabla F_n(\mathbf{x}_t) \right) + (1 - \mathbb{I}_n^t) \left( \nabla \tilde{F}_n(\mathbf{x}_{t,i}^n) - \nabla F_n(\mathbf{x}_{t,i}^n) \right) \right\} \right\|^2 }_{C} \right].$$

$$(16)$$

For term $C$, we have

$$C \overset{(d)}{\le} \frac{\eta}{N^2I} \mathbb{E} \left[ \left\| \sum_{n=1}^{N} \sum_{i=0}^{I-1} \left( \nabla F_n(\mathbf{x}_{t,i}^n) - \nabla F_n(\mathbf{x}_t) \right) \right\|^2 \right]$$

$$+ \frac{\eta}{N^2I} \mathbb{E} \left[ \left\| \sum_{n=1}^{N} \sum_{i=0}^{I-1} (1 - \mathbb{I}_n^t) \left( \nabla \tilde{F}_n(\mathbf{x}_{t,i}^n) - F_n(\mathbf{x}_{t,i}^n) \right) \right\|^2 \right]$$

$$\overset{(e)}{\le} \frac{\eta}{N} \sum_{n=1}^{N} \sum_{i=0}^{I-1} \mathbb{E} \left[ \left\| \left( \nabla F_n(\mathbf{x}_{t,i}^n) - \nabla F_n(\mathbf{x}_t) \right) \right\|^2 \right]$$

$$+ \frac{\eta}{N} \sum_{n=1}^{N} \sum_{i=0}^{I-1} \mathbb{E} \left[ \left\| (1 - \mathbb{I}_n^t) \left( \nabla \tilde{F}_n(\mathbf{x}_{t,i}^n) - \nabla F_n(\mathbf{x}_{t,i}^n) \right) \right\|^2 \right]$$

$$\overset{(f)}{\le} \frac{L^2\eta}{N} \sum_{n=1}^{N} \sum_{i=0}^{I-1} \underbrace{ \mathbb{E} \left[ \left\| \mathbf{x}_{t,i}^n - \mathbf{x}_t \right\|^2 \right] }_{D} + \frac{\eta}{N} \sum_{n=1}^{N} \sum_{i=0}^{I-1} (1 - p_n^t) \mathbb{E} \left[ \left\| \nabla \tilde{F}_n(\mathbf{x}_{t,i}^n) - \nabla F_n(\mathbf{x}_{t,i}^n) \right\|^2 \right],$$

$$(17)$$

where inequality $(d)$ follows from the fact that $\| \sum_{n=1}^{N} \boldsymbol{a}_n \|^2 \le N \sum_{n=1}^{N} \| \boldsymbol{a}_n \|^2$ with $N = 2$, inequality $(e)$ follows from Jensen's inequality, and inequality $(f)$ follows from Assumption 1.

For term $D$, we have

$$D = \mathbb{E} \left[ \left\| \mathbf{x}_{t,i}^n - \mathbf{x}_t \right\|^2 \right]$$

$$= \mathbb{E} \left[ \left\| \mathbf{x}_{t,i-1}^n - \mathbf{x}_t - \eta g_n(\mathbf{x}_{t,i-1}^n) \right\|^2 \right]$$

$$= \mathbb{E} \left[ \left\| \mathbf{x}_{t,i-1}^n - \mathbf{x}_t - \eta \left( g_n(\mathbf{x}_{t,i-1}^n) - \nabla F_n(\mathbf{x}_{t,i-1}^n) + \nabla F_n(\mathbf{x}_{t,i-1}^n) - \nabla F_n(\mathbf{x}_t) \right. \right. \right.$$
$$\left. \left. \left. + \nabla F_n(\mathbf{x}_t) - \nabla f(\mathbf{x}_t) + \nabla f(\mathbf{x}_t) \right) \right\|^2 \right]$$

$$\overset{(g)}{\le} \eta^2 \mathbb{E} \left[ \left\| g_n(\mathbf{x}_{t,i-1}^n) - \nabla F_n(\mathbf{x}_{t,i-1}^n) \right\|^2 \right] + \left( 1 + \frac{1}{2I-1} \right) \mathbb{E} \left[ \left\| \mathbf{x}_{t,i-1}^n - \mathbf{x}_t \right\|^2 \right]$$

$$+ 2I\eta^2 \mathbb{E} \left[ \left\| \nabla F_n(\mathbf{x}_{t,i-1}^n) - \nabla F_n(\mathbf{x}_t) + \nabla F_n(\mathbf{x}_t) - \nabla f(\mathbf{x}_t) + \nabla f(\mathbf{x}_t) \right\|^2 \right]$$

$$\overset{(h)}{\le} \eta^2 \sigma_1^2 + \left( 1 + \frac{1}{2I-1} \right) \mathbb{E} \left[ \left\| \mathbf{x}_{t,i-1}^n - \mathbf{x}_t \right\|^2 \right] + 6I\eta^2 \mathbb{E} \left[ \left\| \nabla F_n(\mathbf{x}_{t,i-1}^n) - \nabla F_n(\mathbf{x}_t) \right\|^2 \right]$$

$$+ 6I\eta^2 \mathbb{E} \left[ \left\| \nabla F_n(\mathbf{x}_t) - \nabla f(\mathbf{x}_t) \right\|^2 \right] + 6I\eta^2 \left\| \nabla f(\mathbf{x}_t) \right\|^2$$

$$\overset{(i)}{\le} \eta^2 \sigma_1^2 + \left( 1 + \frac{1}{2I-1} \right) \mathbb{E} \left[ \left\| \mathbf{x}_{t,i-1}^n - \mathbf{x}_t \right\|^2 \right] + 6L^2I\eta^2 \mathbb{E} \left[ \left\| \mathbf{x}_{t,i-1}^n - \mathbf{x}_t \right\|^2 \right]$$

$$+ 6I\eta^2 \sigma_3^2 + 6I\eta^2 \left\| \nabla f(\mathbf{x}_t) \right\|^2$$

$$= \left( 1 + \frac{1}{2I-1} + 6L^2I\eta^2 \right) \mathbb{E} \left[ \left\| \mathbf{x}_{t,i-1}^n - \mathbf{x}_t \right\|^2 \right] + \eta^2 \sigma_1^2 + 6I\eta^2 \sigma_3^2 + 6I\eta^2 \left\| f(\mathbf{x}_t) \right\|^2, \quad (18)$$

where inequality $(g)$ follows from Young's inequality $\| \mathbf{z_1} + \mathbf{z_2} \|^2 \le (1 + \rho) \| \mathbf{z_1} \|^2 + \left( 1 + \frac{1}{\rho} \right) \| \mathbf{z_2} \|^2$, inequality $(h)$ follows from the fact that $\| \sum_{n=1}^{N} \boldsymbol{a}_n \|^2 \le N \sum_{n=1}^{N} \| \boldsymbol{a}_n \|^2$ with $N = 3$ and Assumption 2, and inequality $(i)$ follows from Assumptions 3 and 4.

Let $\eta \leq \frac{1}{\sqrt{30}LI}$, then we have $\frac{1}{2I-1} + 6L^2 I\eta^2 \leq \frac{1}{I-\frac{1}{2}}$. By unrolling the recursion (Wang & Ji, 2024), we obtain

$$
\mathbb{E}\left[\left\|\mathbf{x}_{t,i}^n - \mathbf{x}_t\right\|^2\right] \leq \sum_{j=0}^{i-1}\left(1 + \frac{1}{I-\frac{1}{2}}\right)^j \left(\eta^2\sigma_1^2 + 6I\eta^2\sigma_3^2 + 6I\eta^2 \|f(\mathbf{x}_t)\|^2\right)
$$

$$
\leq \sum_{j=0}^{I-1}\left(1 + \frac{1}{I-\frac{1}{2}}\right)^j \left(\eta^2\sigma_1^2 + 6I\eta^2\sigma_3^2 + 6I\eta^2 \|f(\mathbf{x}_t)\|^2\right)
$$

$$
= \left[\left(1 + \frac{1}{I-\frac{1}{2}}\right)^I - 1\right]\left(I - \frac{1}{2}\right)\left(\eta^2\sigma_1^2 + 6I\eta^2\sigma_3^2 + 6I\eta^2 \|f(\mathbf{x}_t)\|^2\right)
$$

$$
\overset{(j)}{\leq} 5I\eta^2\left(\sigma_1^2 + 6I\sigma_3^2\right) + 30I^2\eta^2 \|f(\mathbf{x}_t)\|^2, \tag{19}
$$

where equality $(j)$ follows from the fact that

$$
\left[\left(1 + \frac{1}{I-\frac{1}{2}}\right)^I - 1\right]\left(I - \frac{1}{2}\right) = \left[\left(1 + \frac{1}{I-\frac{1}{2}}\right)^{I-\frac{1}{2}}\left(1 + \frac{1}{I-\frac{1}{2}}\right)^{\frac{1}{2}} - 1\right]\left(I - \frac{1}{2}\right)
$$

$$
\leq \left[\sqrt{3}e - 1\right]\left(I - \frac{1}{2}\right) \leq 5I. \tag{20}
$$

Substituting term $D$ into (16), we have

$$
\mathbb{E}\left[f(\mathbf{x}_{t+1})\right] \leq f(\mathbf{x}_t) + \left(\frac{LI\eta^2}{2N^2}\sum_{n=1}^N p_n^t + 5L^2I^2\eta^3\right)\sigma_1^2 + \frac{LI\eta^2\sigma_2^2}{2N^2}\sum_{n=1}^N(1 - p_n^t)
$$

$$
+ 30L^2I^3\eta^3\sigma_3^2 + \eta I\left(30L^2I^2\eta^2 - \frac{1}{2}\right)\|\nabla f(\mathbf{x}_t)\|^2
$$

$$
+ \frac{\eta}{N}\sum_{n=1}^N\sum_{i=0}^{I-1}(1 - p_n^t)\mathbb{E}\left[\left\|\nabla\tilde{F}_n(\mathbf{x}_{t,i}^n) - \nabla F_n(\mathbf{x}_{t,i}^n)\right\|^2\right]. \tag{21}
$$

Let $\eta \leq \frac{1}{2\sqrt{30}LI}$, then we have $30L^2I^2\eta^2 - \frac{1}{2} \leq \frac{1}{4}$. Taking the total expectation of 21, we have

$$
\mathbb{E}\left[\|\nabla f(\mathbf{x}_t)\|^2\right] \leq \frac{4\left(\mathbb{E}\left[f(\mathbf{x}_t) - f(\mathbf{x}_{t+1})\right]\right)}{\eta I} + \left(\frac{2L\eta}{N^2}\sum_{n=1}^N p_n^t + 20L^2I\eta^2\right)\sigma_1^2
$$

$$
+ \frac{2L\eta\sigma_2^2}{N^2}\sum_{n=1}^N(1 - p_n^t) + 120L^2I^2\eta^2\sigma_3^2
$$

$$
+ \frac{4}{NI}\sum_{n=1}^N\sum_{i=0}^{I-1}(1 - p_n^t)\mathbb{E}\left[\left\|\nabla\tilde{F}_n(\mathbf{x}_{t,i}^n) - \nabla F_n(\mathbf{x}_{t,i}^n)\right\|^2\right]. \tag{22}
$$

Then, summing over $T$ rounds and dividing by $T$, we have

$$
\frac{1}{T}\sum_{t=1}^T\mathbb{E}\left[\|\nabla f(\mathbf{x}_t)\|^2\right] \leq \frac{4\left(\mathbb{E}\left[f(\mathbf{x}_0) - f(\mathbf{x}^*)\right]\right)}{\eta IT} + \left(\frac{2L\eta}{TN^2}\sum_{n=1}^N\sum_{t=1}^T p_n^t + 20L^2I\eta^2\right)\sigma_1^2
$$

$$
+ \frac{2L\eta\sigma_2^2}{TN^2}\sum_{t=1}^T\sum_{n=1}^N(1 - p_n^t) + 120L^2I^2\eta^2\sigma_3^2
$$

$$
+ \underbrace{\frac{4}{TNI}\sum_{t=1}^T\sum_{n=1}^N\sum_{i=0}^{I-1}(1 - p_n^t)\mathbb{E}\left[\left\|\nabla\tilde{F}_n(\mathbf{x}_{t,i}^n) - \nabla F_n(\mathbf{x}_{t,i}^n)\right\|^2\right]}_{\epsilon}. \tag{23}
$$

where $\epsilon$ is denoted as the gradient generation error.

To derive the upper bound of $\epsilon$, we introduce the following definition and lemma.

Let the hypothesis class for the gradient generator be

$$\mathcal{H} = \{\mathbf{x} \mapsto f_{\text{GGN}}(g_n(\mathbf{x}); \theta) : \theta \in \Theta\}. \tag{24}$$

For client $n$, define the population risk

$$\ell_n(\theta) := \mathbb{E}_{\mathbf{x}}\left[\|f_{\text{GGN}}(g_n(\mathbf{x}); \theta) - \nabla F_n(\mathbf{x})\|^2\right], \tag{25}$$

and the empirical risk on $r$ i.i.d training samples $\{\mathbf{x}_i\}_{i=1}^r$

$$\hat{\ell}_{n,r}(\theta) := \frac{1}{r}\sum_i^r \|f_{\text{GGN}}(g_n(\mathbf{x}_i); \theta) - \nabla F_n(\mathbf{x}_i)\|^2, \tag{26}$$

with the empirical risk minimization (ERM) estimator

$$\theta_r \in \arg\min_{\theta \in \Theta} \hat{\ell}_{n,r}(\theta), \quad \ell_n^* := \inf_{\theta \in \Theta} \ell_n(\theta). \tag{27}$$

Then we have the following lemma.

**Lemma 3.** *(In-Expectation Learnability). If $\mathcal{H}$ is in-expectation learnable by ERM, then for any $\epsilon_0 > 0$, there exists $r_\mathcal{G}$ such that for all $r \geq r_\mathcal{G}$,*

$$\mathbb{E}[\hat{\ell}_n(\theta_r) - \ell_n^*] \leq \epsilon_0(r). \tag{28}$$

Equivalently,

$$\mathbb{E}_{\mathbf{x}}\left[\|f_{\text{GGN}}(g_n(\mathbf{x}); \theta) - \nabla F_n(\mathbf{x})\|^2\right] \leq \underbrace{\inf_\theta \mathbb{E}_{\mathbf{x}}\left[\|f_{\text{GGN}}(g_n(\mathbf{x}); \theta) - \nabla F_n(\mathbf{x})\|^2\right]}_{:=\epsilon_n^*} + \epsilon_0(r). \tag{29}$$

Lemma 3 decomposes the surrogate-gradient error into an approximation term $\epsilon_n^*$ due to the capacity of the generator class $\mathcal{H}$, and a statistical term $\epsilon_0(r)$ that shrinks with the effective sample size $r$. Consequently, once the GGN is trained with sufficiently many samples, every occurrence of $\mathbb{E}\left[\left\|\nabla \tilde{F}_n(\mathbf{x}_{t,i}^n) - \nabla F_n(\mathbf{x}_{t,i}^n)\right\|^2\right]$ in the convergence proof can be replaced by $\epsilon_{n,t}^* + o(1)$; the residual $o(1)$ scales as $\mathcal{O}(1/\sqrt{r})$ under standard ERM rates.

Recall the term $\epsilon$ in (23) and by Lemma 3, we have

$$\mathbb{E}\left[\left\|\nabla \tilde{F}_n(\mathbf{x}_{t,i}^n) - \nabla F_n(\mathbf{x}_{t,i}^n)\right\|^2\right] \leq \epsilon_{n,t}^* + \epsilon_0(r_{n,t}), \tag{30}$$

where $r_{n,t}$ is is the effective number of samples that trained the GGN used at $(n,t)$, and $\epsilon_0(r_{n,t})$ is the ERM excess-risk term from Lemma 3. Then according to (Mey, 2022; Nair et al., 2025; Gu et al., 2021), we take the following assumption.

**Assumption 5.** *(Standard Rate). Under square-loss regression with bounded or sub-Gaussian noise and a Lipschitz hypothesis class, ERM satisfies*

$$\epsilon_0(r_n) \leq \frac{C_1}{\sqrt{r}} \quad \textit{for some constant } C_1 > 0. \tag{31}$$

Let $\tau_n^t$ denote the gradient staleness used for the generated update at $(n,t)$, a mild requirement is that the effective sample size grows at least inversely with staleness, e.g.

$$r_{n,t} \gtrsim \frac{t}{\tau_n^t} \implies \epsilon_0(r_{n,t}) \leq C_1\sqrt{\frac{\tau_n^t}{t}}. \tag{32}$$

Then we can bound term $\epsilon$ as follows.

$$\epsilon = \frac{4}{TNI}\sum_{t=1}^T\sum_{n=1}^N\sum_{i=0}^{I-1}(1-p_n^t)\mathbb{E}\left[\left\|\nabla \tilde{F}_n(\mathbf{x}_{t,i}^n) - \nabla F_n(\mathbf{x}_{t,i}^n)\right\|^2\right]$$

$$\leq \frac{4}{TNI} \sum_{t=1}^{T} \sum_{n=1}^{N} \sum_{i=0}^{I-1} (1 - p_n^t) \left( \epsilon_{n,t}^* + C_1 \sqrt{\frac{\tau_n^t}{t}} \right)$$

$$\leq \frac{4}{TN} \sum_{t=1}^{T} \sum_{n=1}^{N} (1 - p_n^t) \epsilon_{n,t}^* + \frac{4C_1}{TN} \sum_{t=1}^{T} \sum_{n=1}^{N} (1 - p_n^t) \frac{\sqrt{\tau_n^t}}{\sqrt{t}}$$

$$\overset{(j)}{\leq} \frac{4}{TN} \sum_{t=1}^{T} \sum_{n=1}^{N} (1 - p_n^t) \epsilon_{n,t}^* + \frac{4C_1}{T} \sum_{t=1}^{T} \frac{1}{\sqrt{t}} \sqrt{\frac{1}{N} \sum_{n=1}^{N} (1 - p_{n,\min})} \sqrt{\frac{1}{N} \sum_{n=1}^{N} \tau_{n,\max}}$$

$$\overset{(k)}{\leq} \frac{4}{TN} \sum_{t=1}^{T} \sum_{n=1}^{N} (1 - p_n^t) \epsilon_{n,t}^* + \frac{8C_1 \sqrt{\overline{\tau}_{\max}(1 - \overline{p}_{\min})}}{\sqrt{T}}. \tag{33}$$

where inequality $(j)$ follows from the Cauchy-Schwarz inequality, and $\tau_{n,\max} = \max\{\tau_n^t\}, p_{n,\min} = \min\{p_n^t\}, \forall t$, and inequality $(k)$ follows from the fact that $\sum_{t=1}^{T} \frac{1}{\sqrt{t}} \leq 2\sqrt{T}$, and $\overline{\tau}_{\max} = \frac{1}{N} \sum_{n=1}^{N} \tau_{n,\max}, \overline{p}_{\min} = \frac{1}{N} \sum_{n=1}^{N} p_{n,\min}$.

Let $\eta = \min\{\frac{1}{\sqrt{IT}}, \frac{1}{2\sqrt{30}LI}\}$, we have

$$\frac{1}{T} \sum_{t=1}^{T} \mathbb{E}\left[ \|\nabla f(\mathbf{x_t})\|^2 \right]$$

$$\leq \mathcal{O}\left( \frac{(\mathbb{E}[f(\mathbf{x_0}) - f(\mathbf{x^*})])}{\sqrt{IT}} + \frac{L(\sigma_1^2 \overline{p} + \sigma_2^2 \overline{q})}{N\sqrt{IT}} + \frac{L^2(\sigma_1^2 + I\sigma_3^2)}{T} + \overline{\epsilon}^* + \frac{\sqrt{\overline{\tau}_{\max}(1 - \overline{p}_{\min})}}{\sqrt{T}} \right), \tag{34}$$

where $\overline{p} = \frac{1}{TN} \sum_{t=1}^{T} \sum_{n=1}^{N} p_n^t, \overline{q} = 1 - \overline{p}$, and $\overline{\epsilon}^* = \frac{1}{TN} \sum_{t=1}^{T} \sum_{n=1}^{N} (1 - p_n^t) \epsilon_{n,t}^*$.

Thus, we complete the proof.

## A.2 PROOFS OF THEOREM A.2

We now specialize to a simple yet informative setting in which each device participates independently in each round with a fixed probability $p_n^t = p_n \in (0, 1], \forall t$. In this case, we have

$$\mathbb{E}[\tau_n^{t+1}] = (1 - p_n)(\mathbb{E}[\tau_n^t] + 1). \tag{35}$$

Summing over $t$ rounds, we obtain

$$\mathbb{E}[\tau_n^t] = \frac{1 - p_n}{p_n}(1 - (1 - p_n)^{t-1}) \leq \frac{1 - p_n}{p_n}. \tag{36}$$

By Jensen's inequality, we have

$$\mathbb{E}[\sqrt{\tau_n^t}] \leq \sqrt{\mathbb{E}[\tau_n^t]} \leq \sqrt{\frac{1 - p_n}{p_n}}. \tag{37}$$

Incorporating (37) into term $\epsilon$, applying $\sum_{t=1}^{T} \frac{1}{\sqrt{t}} \leq 2\sqrt{T}$, and taking the expectation of $\tau_n^t$ over $t$, we obtain

$$\epsilon = \frac{4}{TNI} \sum_{t=1}^{T} \sum_{n=1}^{N} \sum_{i=0}^{I-1} (1 - p_n) \mathbb{E}\left[ \left\| \nabla \tilde{F}_n(\mathbf{x_{t,i}^n}) - \nabla F_n(\mathbf{x_{t,i}^n}) \right\|^2 \right]$$

$$\leq \frac{4}{TNI} \sum_{t=1}^{T} \sum_{n=1}^{N} \sum_{i=0}^{I-1} (1 - p_n) \left( \epsilon_{n,t}^* + \mathbb{E}\left[ C_1 \sqrt{\frac{\tau_n^t}{t}} \right] \right)$$

$$\leq \frac{4}{TN} \sum_{t=1}^{T} \sum_{n=1}^{N} (1 - p_n) \epsilon_{n,t}^* + \frac{4C_1}{TN} \sum_{n=1}^{N} (1 - p_n) \sum_{t=1}^{T} \frac{\mathbb{E}[\sqrt{\tau_n^t}]}{\sqrt{t}}$$

$$\leq \frac{4}{TN} \sum_{t=1}^{T} \sum_{n=1}^{N} (1-p_n) \epsilon_{n,t}^* + \frac{8C_1}{\sqrt{T}N} \sum_{n=1}^{N} \frac{(1-p_n)^{\frac{3}{2}}}{\sqrt{p_n}}$$

$$\leq \frac{4}{TN} \sum_{t=1}^{T} \sum_{n=1}^{N} (1-p_n) \epsilon_{n,t}^* + \frac{8C_1}{\sqrt{T}} \frac{(1-p_{\min})^{\frac{3}{2}}}{\sqrt{p_{\min}}}. \tag{38}$$

Then we have

$$\frac{1}{T} \sum_{t=1}^{T} \mathbb{E}\left[\|\nabla f(\mathbf{x_t})\|^2\right]$$

$$\leq \mathcal{O}\left( \frac{(\mathbb{E}\left[f(\mathbf{x_0}) - f(\mathbf{x^*})\right])}{\sqrt{IT}} + \frac{L(\sigma_1^2 \bar{p} + \sigma_2^2 \bar{q})}{N\sqrt{IT}} + \frac{L^2(\sigma_1^2 + I\sigma_3^2)}{T} + \bar{\epsilon}^* + \frac{(1-p_{\min})^{\frac{3}{2}}}{\sqrt{T}p_{\min}} \right). \tag{39}$$

Thus, we complete the proof.

