# OpenReview forum: "LOGIT: Learning to Generate Gradients for Federated Learning with Arbitrary Client Unavailability"
_ICLR.cc/2026/Conference — Submitted to ICLR 2026_

### Official Review · Reviewer_PCbU · 2025-10-28

**Soundness:** 1
**Presentation:** 3
**Contribution:** 1
**Rating:** 2
**Confidence:** 4

**Summary:**

This paper proposes **LOGIT**, a gradient-generation framework for federated learning (FL) under arbitrary client unavailability. The method aims to reconstruct missing client gradients on the server by learning client-specific gradient trajectories conditioned on historical and current updates from available clients. The authors claim that LOGIT enables unbiased surrogate gradients for dropped clients, derives a tighter convergence bound with rate $\mathcal{O}(1/\sqrt{T})$, and empirically outperforms existing methods on public datasets.

**Strengths:**

The paper trys to address an important and practical problem in federated learning, namely client unavailability. The idea of reconstructing unavailable gradients using a generator network represents an interesting attempt to improve learning continuity. The paper is also clearly written and well-organized, making the methodology easy to understand.

**Weaknesses:**

The proposed methodology requires the server to collect gradients from individual clients, which makes it incompatible with secure aggregation and differential privacy mechanisms, as both rely on hiding individual client updates. Moreover, the approach assumes that each client’s gradient trajectory can be effectively learned, but in practice, gradients tend to diminish and fluctuate dynamically during training. It is unclear how the proposed GCN-based generator can reliably capture and predict such evolving gradient patterns.

In addition, the data available for training each client’s GCN is extremely limited. For example, in a conventional federated learning system with 100 clients, where 10 clients participate per round and the total number of rounds is 100, the server can only obtain about 10 gradient samples per client. Under such conditions, it is questionable whether the GCN can learn meaningful gradient dynamics or generalize well. This may also require significantly more communication rounds, further increasing the training cost.

In the main experiments, the number of clients is set to 10, which is too small to validate the effectiveness of the proposed algorithm.

The methodology also lacks scalability, as the complexity of training the GCN grows linearly with the number of clients, and the model size may need to increase with the dimensionality of the gradients.

With Assumption 3, the analysis becomes nearly identical to that of conventional FedAvg.

Overall, the GCN component appears heuristic and insufficiently justified.

**Questions:**

What's the reason for you to choose GCN as the generator?

According to existing studies on FedAvg, this algorithm is already quite robust to partial client participation. Therefore, it is unclear whether it is truly necessary to recover the gradients of each client using such a heuristic methodology.

Given the limited number of communication rounds typically available in federated learning, it is doubtful that one can reliably identify the dynamic gradient patterns of individual clients. The feasibility of accurately modeling these trajectories under practical constraints remains highly questionable.

---

### Official Review · Reviewer_1uDW · 2025-10-30

**Soundness:** 2
**Presentation:** 2
**Contribution:** 2
**Rating:** 2
**Confidence:** 4

**Summary:**

This paper proposes LOGIT, a framework to address client dropouts in Federated Learning (FL) by generating surrogate gradients. A server-side Gradient Generation Network (GGN) is trained for each client. When a client is unavailable, its GGN generates a substitute gradient based on its own gradient history and information from currently available clients.

**Strengths:**

* The idea of using a generative model to create surrogate gradients for missing clients is novel and provides a new perspective on tackling the client unavailability problem.
* The paper provides a complete convergence analysis, establishing a rate of O(1/√T) and offering insights into how factors like participation probability and gradient staleness affect convergence.
* The method is validated on three public datasets (CIFAR-10, CIFAR-100, IMAGENETTE), which helps demonstrate its effectiveness across different tasks.

**Weaknesses:**

* There are minor typos in the paper, such as the repeated "denote denote" and the redundant sentence "the cosine similarity. denotes the cosine similarity." on page 4.
* The server is required to store the most recent gradient for each client. This could lead to prohibitive memory overhead in large-scale FL scenarios with many clients, as each stored gradient has the same size as the global model.
* The main experiments are conducted with only N=10 clients, which may not be representative of large-scale FL. It is also unclear if the experiments assume all 10 clients are selected each round (full participation) before dropouts occur. The performance under a more common partial participation setting is not explored.
* The client availability of p=0.5 in the main experiments is an extreme setting. Figure 3(a) shows that the performance difference between p=0.5 and p=0.9 is not substantial, which might suggest that the impact of client dropouts is not as significant as claimed, and the proposed method's benefits may be marginal in more realistic scenarios.
* The baselines are limited to FedAvg and two methods based on reusing stale gradients (MIFA, WS). The paper should include comparisons with stronger baselines mentioned in the related work, such as FedProx or SCAFFOLD, which are designed to handle statistical heterogeneity and may inherently be more robust to biased aggregation from dropouts.

**Questions:**

* According to Algorithm 1, the GGN is trained and used online from the very beginning. In the early rounds, the GGN is not well-trained and may generate poor-quality gradients, which could be detrimental to the main model's convergence. Have the authors investigated the effect of this "cold start" problem?
* The convergence analysis seems to assume a bounded error for the GGN. Does this analysis fully account for the nested optimization process where the GGN is being trained online on a non-stationary target, as the client gradients change while the global model evolves?
* Could the authors provide learning curves (accuracy vs. communication rounds) for the main experiments? This would offer a clearer view of the training dynamics and convergence speed compared to just reporting the final accuracy or rounds to target.
*  Regarding the "Client-specific coordinatewise GGN," the paper states the same small network is applied to every coordinate. Does this mean for each client n, there is one single small network Φ(·; θn) that processes each of the d gradient coordinates independently? Even if so, the server compute of O(Kd) could still be substantial for models with very high dimensionality (d). Is it practically feasible for very large models?

---

### Official Review · Reviewer_rUQc · 2025-11-03

**Soundness:** 3
**Presentation:** 2
**Contribution:** 3
**Rating:** 4
**Confidence:** 2

**Summary:**

This paper processes LOGIT, a novel framework for FL that addresses client unavailability caused by communication failures or dropouts. LOGIT introduces a server-side Gradient Generation Network (GGN), which learns client-specific gradients from historical updates and current active clients to generate surrogate gradients for missing clients. This LOGIT shows a superior accuracy and communication efficiency across multiple datasets.

**Strengths:**

1. This paper addresses a very practical issue in FL, which is the possibility of broken communication between the server and clients that results in missing gradient updates in some rounds. By focusing on this intermittent connectivity problem, the paper tackles a common real-world challenge that can bias gradient aggregation and impede convergence in FL.
2. The authors design a comprehensive set of ablation studies to evaluate LOGIT under various realistic conditions. In particular, they systematically vary the communication delay, the fraction of successful communications, and the number of participating clients. By examining these factors, the experiments convincingly demonstrate the effectiveness and robustness of LOGIT from multiple perspectives, highlighting that it can preserve model performance even when communication is delayed or sporadic.

**Weaknesses:**

1. It is unclear how accurate the surrogate gradient generator in LOGIT is over the course of training. The paper would benefit from an analysis of whether the generator's outputs become more precise as training progresses. For instance, monitoring the average error between the actual gradients and the generated surrogate gradients would illustrate the generator's fidelity, Additionally, since the surrogate gradients are likely of lower quality in the eraly traiing phase, the authors could clarify if any techniques (e.g., a warm-up period or special training schedule) are used to mitigate the impact of initially inaccurate surrogate updates.
2. The generator's design seems misaligned with the goal of accurate gradient prediction. According to Eq. 3, the loss function $L_{MSE}$ focuses the generator's output to be close to the input gradient. In other words, the current design encourages the generator to reproduce the most recent stable gradient rather than predicting the unseen new gradient. This raises concerns that LOGIT's generator might be merely outputting a slightly adjusted version of the previous gradients, instead of leveraging the client's gradient trajectory to forecast the missing update (as a sequential model would). Clarification on how the generator accounts for temporal patterns would strengthen the work.
3. The paper does not clearly explain how $\lambda$ is chosen in Eq. 3 to balance the two loss objectives. Is this $\lambda$ fixed or tuned for each client, and does its optimal value depend on the degree of data heterogeneity across clients? Moreover, the formulation of the generator's loss as $L_{MSE} - L_{Align}$ is unusual, since subtracting $L_{Align}$ is equivalent to maximizing the alignment term. The authors should clarify the rationale behind subtracting this $L_{Align}$ rather than adding it.
4. Several notational and presentation issues could be addressed to improve clarity.
- L 223, the term "client $n'$" appears without definition.
- L 195, the notation $g_n(x_t), k \in  \mathcal{A}_t  \backslash  \{n\}$ is confusing, since $k$ is not defined in $g_n(x_t)$.
- L 244, the summation is from $i=1$ to $i=I-1$ for $x_{t, i-1}^n$, whereas Algorithm 1 L8 suggests it should be $i=0$ to $i=I-1$ for $x_{t, i}^n$.
- L 364, the set notation is missing braces.
- It would be helpful to unify and clearly explain the notation for superscript and subscript.

**Questions:**

Refer to the weaknesses section.

---

### Official Review · Reviewer_oTnz · 2025-11-05

**Soundness:** 2
**Presentation:** 4
**Contribution:** 3
**Rating:** 2
**Confidence:** 4

**Summary:**

This paper proposes a novel framework for federated learning with temporarily unavailable clients.
In the proposed approach, the gradients that are missing from clients are reconstructed based on the previously observed gradients from this client and the gradients observed from other clients.
A theoretical analysis is performed, showing a $O(1/\sqrt{T})$ convergence rate, and scaling with the "gradient staleness" (i.e., maximal time between reception of consecutive gradients from a client).
To reduce the complexity of the proposed method, it is applied to the gradient coordiante-per-coordinate, avoiding the need to consider correlations between coordinates.
Experimental results show the relevance of the method on multiple classical problems, demonstrating the the proposed method exhibits improved performance in comparison with other existing methods.

**Strengths:**

1. The idea of generating missing gradients in cases where clients are unavailable is interesting and, to my knowledge, original.
2. The proposed methods exhibits strong empirical results in the case where clients are unavailable following a Bernoulli distribution, outperforming other baselines while reducing the number of communications.
3. Theoretical convergence guarantees are provided, proving that the method converges despite gradient staleness, although (as expected) the time between successive computation of the gradients slows down the training.

**Weaknesses:**

1. Assumption 3, that the gradient computed by the proposed algorithm is unbiased, seems overly strong. In my understanding, this essentially assumes that the model used for generating gradient is unbiased, which is very likely not the case given the setting used for generation.
2. While the coordinate-wise scheme for gradient generation significantly reduces the computational cost, it may create strong bias in the generated gradients, which may prevent convegence in complex scenarios.
3. To estimate unavailable models, the server needs to store the previously received gradient for every client, which may have a prohibitive cost when the number of clients is large.
4. Experiments consider a limited setting where clients participate following a fixed Bernoulli distribution: this is a restrictive framework, where clients have a fixed, pre-determined behaviour, and contrasts with the claim that the method work in very general settings.

**Questions:**

1. Does Assumption 3 indeed mean that the gradient generation algorithm is unbiased? If so, is it possible to replace this assumption by a milder one, allowing to prove that the generation process itself is unbiased?
2. It seems that the loss function proposed for generating the gradients in Eq. (3) depends on the iteration count $t$ through the sampled client set $\mathcal{A}_t$: this seems contradictory with the fact that the loss $\mathcal{L}_{GNN}$ does not depend on $t$: is this loss really the considered loss?
3. In 3.3, authors claim that "gradients are $\ell_2$-normalized when fed to the GCN, while raw when aggregated in fedavg: what is the rational behind this? Doesn't it result in inconsistent gradient updates?

There are a few typos:
- in 3.1: the variant of FedAvg that is presented seems to be using a single local step, while the opposite is claimed
- in Eq. (3), $\mathcal{L}_{GNN}$ seems to be $\mathcal{L}_{GCN}$: if not, why call it "GNN"?

---

### Meta-Review · Area_Chair_9etP · 2025-12-31

**Summary:**

This paper proposed LOGIT: Learning to Generate Gradients for Federated Learning with Arbitrary Client Unavailability.

There is a general consensus on rejection from four reviewers, and the authors did not respond.

Reviewers acknowledge the problem is interesting, and the motivation is important. Some reviewers also mention the novelty of using a generator network to generate missing gradients.

However, reviewers raised concerns about the restrictive (and likely unrealistic) assumption for the theory; the limited setting of Bernoulli distribution in simulation; the large cost of the algorithm tracking gradients for all clients; and the lack of analysis for important generator design points.

**Reviewer Concerns:**

There is no rebuttal from authors. Reviewers' concerns are not addressed.

**Reviewer Scores:**

No discussion, and hence no score changes.

---

### Decision · Program_Chairs · 2026-01-26

Reject